# Evidence of Placental Aging in Late SGA, Fetal Growth Restriction and Stillbirth—A Systematic Review

**DOI:** 10.3390/biomedicines11071785

**Published:** 2023-06-21

**Authors:** Anna Kajdy, Dorota Sys, Jan Modzelewski, Joanna Bogusławska, Aneta Cymbaluk-Płoska, Ewa Kwiatkowska, Magdalena Bednarek-Jędrzejek, Dariusz Borowski, Katarzyna Stefańska, Michał Rabijewski, Arkadiusz Baran, Andrzej Torbe, Stepan Feduniw, Sebastian Kwiatkowski

**Affiliations:** 1First Department of Obstetrics and Gynecology, Centre of Postgraduate Medical Education, 01-004 Warsaw, Poland; jmodzelewski@cmkp.edu.pl (J.M.); abaran@cmkp.edu.pl (A.B.); 2Department of Medical Statistics, School of Public Health, Centre of Postgraduate Medical Education, 01-828 Warsaw, Poland; dsys@cmkp.edu.pl; 3Department of Biochemistry and Molecular Biology, Centre of Postgraduate Medical Education, 01-813 Warsaw, Poland; joanna.boguslawksa@cmkp.edu.pl; 4Department of Gynecological Surgery and Gynecological Oncology of Adults and Adolescents, Pomeranian Medical University, 70-111 Szczecin, Poland; aneta.cymbaluk@gmail.com; 5Department of Nephrology, Transplantology and Internal Medicine, Pomeranian Medical University, 70-111 Szczecin, Poland; ewakwiat@gmail.com; 6Department Obstetrics and Gynecology, Pomeranian Medical University, 70-111 Szczecin, Poland; m.bednarekjedrzejek@gmail.com (M.B.-J.); torbea@wp.pl (A.T.); kwiatkowskiseba@gmail.com (S.K.); 7Clinic of Obstetrics and Gynaecology, Provincial Combined Hospital in Kielce, 25-736 Kielce, Poland; darekborowski@gmail.com; 8Department of Obstetrics, Medical University of Gdańsk, 80-210 Gdańsk, Poland; kciach@wp.pl; 9Department of Reproductive Health, Center of Postgraduate Medical Education, Centre of Postgraduate Medical Education, Żelazna 90 St., 01-004 Warsaw, Poland; mirab@cmkp.edu.pl; 10Department of Gynecology, University Hospital Zürich, 8091 Zürich, Switzerland; stepan.feduniw@gmail.com

**Keywords:** pregnancy, obstetric complications, placental aging, cellular senescence, late FGR, late SGA, stillbirth

## Abstract

During pregnancy, the placenta undergoes a natural aging process, which is considered normal. However, it has been hypothesized that an abnormally accelerated and premature aging of the placenta may contribute to placenta-related health issues. Placental senescence has been linked to several obstetric complications, including abnormal fetal growth, preeclampsia, preterm birth, and stillbirth, with stillbirth being the most challenging. A systematic search was conducted on Pubmed, Embase, and Scopus databases. Twenty-two full-text articles were identified for the final synthesis. Of these, 15 presented original research and 7 presented narrative reviews. There is a paucity of evidence in the literature on the role of placental aging in late small for gestational age (SGA), fetal growth restriction (FGR), and stillbirth. For future research, guidelines for both planning and reporting research must be implemented. The inclusion criteria should include clear differentiation between early and late SGA and FGR. As for stillbirths, only those with no other known cause of stillbirth should be included in the studies. This means excluding stillbirths due to congenital defects, infections, placental abruption, and maternal conditions affecting feto-maternal hemodynamics.

## 1. Introduction

As pregnancy progresses, the placenta undergoes a natural process of aging, which is considered a physiological occurrence. It has been hypothesized that an abnormally accelerated and premature aging of the placenta may contribute to placenta-related health issues [1]. Placental senescence has been identified as a pathological factor that can lead to various obstetric complications, including abnormal fetal growth, preeclampsia, preterm birth, and stillbirth. The most difficult of these complications is late fetal growth restriction and term fetal death, which can result in stillbirth [2,3]. Abnormal growth in the third trimester is classified into two types: small for gestational age (SGA) fetuses and fetal growth restriction (FGR) [4]. One differs from the other by the presence of signs of placental insufficiency, which include estimated fetal weight below the third percentile, weight between the third and tenth centile, or weight crossing centiles but with the presence of abnormal Doppler parameters [5]. SGA and FGR both carry an increased risk of stillbirth, but most term deaths are found in normally grown fetuses [6]. The majority of SGA neonates are born at term, but fetuses with estimated weight below the tenth centile are 2–3 times more often found among stillbirths [6]. Studies have shown a complex association between small size and stillbirth, as stillborn neonates tend to lose around 20–25% of their body weight while in the uterus [7]. Although being small is considered a risk factor for stillbirth, the majority of stillbirths that occur at term involve fetuses that have grown normally. Only 30–40% of stillbirths that occur after 32 weeks of gestation are classified as being below the tenth percentile for fetal size [8,9,10,11]. For this reason, despite different management strategies for term pregnancies, late stillbirth remains a challenge of perinatal medicine [8,12].

At term, most stillbirths occur in fetuses that have grown normally and are classified as “unexplained,” which often leads to the assumption that these events are unpreventable [11,13], although it has been hypothesized that underlying placental pathology is the key to understanding these deaths [8]. Unfortunately, not all symptoms present early enough to both the mother and medical provider for early intervention to be made. By definition, fetal growth restriction is not achieving programmed growth potential [14,15]. There is growing evidence that a proportion of appropriate for gestational age (AGA) neonates are indeed growth restricted [16]. This correlation could help decipher “unexplained” late stillbirths.

Aging is primarily characterized by a gradual decline in cellular, tissue, and organ function, resulting in the accumulation of senescent cells in mitotic tissues. The placenta, which is programmed to function for the duration of the pregnancy, also undergoes these processes [17]. These cells, which have started the aging process, disrupt the normal function of tissues by affecting neighboring cells, breaking down the extracellular matrix, and reducing the tissues’ regenerative capacity. This decline is due to a reduction in the number of stem and progenitor cells [18].

At the morphological level, senescence is associated with several characteristic features. These include cellular and tissue atrophy, reduced cell proliferation, altered cell morphology, and the accumulation of cellular debris. In many tissues, such as the skin, there is a decline in the number and activity of specialized cells such as fibroblasts and melanocytes, leading to the appearance of wrinkles, thinning, and loss of elasticity [17,18].

At the molecular level, senescence involves various mechanisms, including alterations in gene expression, telomere shortening, genomic instability, epigenetic modifications, mitochondrial dysfunction, and cellular senescence-associated secretory phenotype (SASP). Hormones play a crucial role in modulating these processes and influencing the pace of senescence [19].

One of the well-studied hormonal systems relating to senescence is the hypothalamic–pituitary–adrenal (HPA) axis. The HPA axis regulates the production and release of cortisol, a hormone involved in stress response. With age, the HPA axis becomes dysregulated, leading to altered cortisol levels. Elevated cortisol levels can accelerate the aging process, affecting multiple organ systems and contributing to the development of age-related diseases [20,21].

Fetal growth restriction and its consequences, including fetal death, varies in severity depending on the degree of abnormal placental development. It has been hypothesized that the changes observed in late placentation abnormalities may not be as significant, but the role of maternal morbidity and environmental stressors cannot be ignored. These stressors may trigger early placental senescence, contributing to the development of these pregnancy complications.

This review aimed to present the current understanding of the role of placental aging in late SGA, fetal growth restriction, and term stillbirth, and identify future research directions in this area.

## 2. Materials and Methods

This systematic review is registered in OSF Registries (doi.org/10.17605/OSF.IO/SJ93D). It was conducted according to the Preferred Reporting Items for Systematic Reviews and Meta-Analyses (PRISMA) Statement guidelines [22].

### 2.1. Search Strategy

Medline, Web of Science, Cochrane, Embase, and Scopus were searched using the search strategy presented in Table 1.

### 2.2. Inclusion and Exclusion Criteria

No time limits were set. All peer-reviewed types of publications in English were included, except conference abstracts.

The original research was limited to studies conducted on placentas from pregnancies complicated by late SGA, FGR, or stillbirth. Review articles were limited to those that presented data relevant to pregnancies complicated by late SGA, FGR, or stillbirth.

### 2.3. Study Selection

Sixty full-text articles were assessed for eligibility. The study titles and abstracts were screened according to the following inclusion/exclusion criteria. The references cited in the found articles were also searched in order to identify other published articles on the topic. The study selection process is depicted in the PRISMA flow chart (Figure 1).

### 2.4. Process of Data Collection

The search was performed on 10 January 2023. The retrieved records were downloaded and imported into Zotero’s virtual library, which was accessed by the evaluation researchers. After the selection of articles for full-text analysis, they were also deposited in Zotero. The process of synthesizing the results was done using text documents stored in protected files in the virtual workspace. The study was completed on 31 March 2023.

### 2.5. Evaluation of Risk of Bias

The Newcastle—Ottawa quality assessment scale was used to assess the risk of bias in all the original research [23]; for observational studies, two reviewers independently assessed the risk of bias in each study using the aforementioned scale. The SANRA scale was used for the quality assessment of narrative review articles [24].

### 2.6. Synthesis Methods

The most important data from the publication were extracted and presented in a systematic approach; see Table 2 and Table 3. For the original research, four columns were developed; these included the publication author(s), publication year, population, and key points. The key points column presents relevant information on late SGA, FGR, and stillbirth from the research. For the narrative reviews, a parallel table was prepared that included the publication author(s), publication year, type of review, SANRA quality assessment, and key points. The data in this review were synthesized using thematic analysis and grouping similar information.

## 3. Results

### 3.1. Characteristics of the Studies

For the final synthesis, we identified 22 full-text articles. Of these, 15 presented original research and 7 presented narrative reviews.

None of the narrative reviews specifically synthesized knowledge regarding late SGA, FGR, and stillbirth. Only one review concentrated solely on the role of placental aging in unexplained stillbirths at term. The review articles scored between 6 and 10 points in the qualitative assessment of narrative reviews. They primarily lacked justification for performing the review, did not state the hypothesis, and failed to describe the search strategies. Six original studies received the highest possible quality rating. The remaining original research studies presented several sources of bias (Table 4). First of all, there were no unified definitions of SGA, FGR, or stillbirth. The studies had mixed early and late FGR groups. We included studies with a subgroup of pregnancies with an average gestational age above 32 weeks, but this was not always easy to identify from the presented demographic data. The control groups were not always well described. Indeed, the control groups were matched by gestational age, but the reasons for preterm delivery were not specified. As for stillbirths, none of the studies described exclusion criteria for the known causes of stillbirths or the relationship of stillbirth with birthweight (SGA, AGA, LGA). The list of articles and the key points of the presented research are described in Table 2 and Table 3. This has made comparison of the results and synthetical analyses very difficult.

### 3.2. Synthesis of the Results

The following areas of research were identified in the original studies. Placental AURK expression decreases with gestational age, and AURKC is reduced in the placentas from pregnancies complicated by severe early-onset FGR. Additionally, the mtDNA copy number is increased in FGR and SGA placentas, and telomere length attrition is associated with stillbirth. Placentas from FGR cases exhibit accelerated aging, decreased telomerase activity, and shorter telomeres. FGR is associated with decreased activity of mTORC1 and mTORC2, increased expression of *P53* mRNA and protein, and increased apoptosis (Figure 1). Furthermore, alpha klotho levels in cord blood are lower in cases of SGA, indicating a potential link between alpha klotho and the accelerated maturation of placental villi. These findings suggest that alterations in placental gene expression, mitochondrial function, telomere length, and protein expression may contribute to FGR and placental aging.

The review articles highlighted various mechanisms that may contribute to abnormal growth. Telomere erosion and cellular senescence in the placenta were found to contribute to FGR, with shorter telomeres observed in FGR placentas compared to controls. Placental mTOR is blocked when maternal folate concentrations are low, leading to decreased placental amino acid transport and fetal nutrient unavailability. FGR coexists with increased expression of DNA damage biomarkers, reduction of telomere length and telomerase activity, upregulation of senescence-associated markers, and oxidative DNA damage. Placental apoptosis may result from placental hypoxia, reactive oxygen species, or a reduction of growth factors, and understanding cell turnover pathways may provide a novel therapeutic approach. Finally, changes in the late gestation placenta contribute to unexplained antepartum stillbirths, and genes that produce aging affect fewer pregnancies, with polymorphisms in genes that produce these effects remaining in the population (Figure 2).

## 4. Discussion

The presented research is the first that addresses the problem of late SGA, FGR, and stillbirth in a systematic review. SGA and FGR are two primary forms of prenatally diagnosed abnormal growth [3]. FGR is characterized by alterations in fetoplacental Doppler and poses a greater risk of intrauterine deterioration and mortality compared to SGA. SGA fetuses are often referred to as constitutionally small and have a near-normal perinatal outcome [3,4]. Both SGA and FGR are associated with suboptimal neurodevelopmental outcomes and intrauterine cardiovascular programming [44]. Placental dysfunction has been most commonly associated with abnormal fetal growth, but only 25% of pregnancies complicated by FGR show abnormalities on histopathological examinations in late forms [3]. For this reason, new approaches need to be researched to detect subtle changes in placental development.

An interesting hypothesis has been made by Smith et al. in their review of placental aging in unexplained stillbirth [42]. These authors refer to Guarente’s definition of aging “as the increase in the likelihood of death occurring with advancing time.” Of course, this definition is meant for mature individuals. Still, suppose we treated a developed placenta as an entity with a programmed life span. In this case, the definition brings new light to the discussion [45]. Analysis of the risk of stillbirths has numerously linked it to gestational age, meaning the older the pregnancy (meaning placenta), the higher the risk of unexplained intrauterine death of an otherwise healthy fetus [46]. Smith et al. hypothesize that etiologies associated with unexplained intrauterine death, such as infarction, hemorrhagic, or placental thrombotic events, are indeed associated with placental aging, as is grown-up human death from cardiac or cerebrovascular disease [42]. Similarly, in late SGA and FGR, exposure to stressors after completed implantation could result in progressive placental aging, abnormal growth, subtle adaptive changes, death, or intrauterine programming, resulting in a higher risk of abnormal development, and metabolic and cardiovascular diseases [3,41,44,47].

Since aging is a physiological process, the key in this discussion is to differentiate physiological from pathological cellular senescence and placental aging. The senescence of trophoblast cells is expected to progress with normally advancing pregnancy and, therefore, results in placental aging. Cellular senescence markers, including SA-ß-gal, the increased expression of p16 and p21 (CDK inhibitors), and p53 (tumor suppressor), have been identified in the syncytiotrophoblast of full-term pregnancies in various studies [1,2,41]. p16, 21, and 53 are responsible for keeping pRB (retinoma tumor suppressor protein) in its active state [33]. pRB suppresses transcription factors that are essential for cell proliferation [1,2,41]. The progression of the G1 and S phases of the cell cycle is controlled by *E2F* target genes. Silencing of *E2F* target genes by pRB results in the accumulation of reorganized heterochromatin structures in senescent cell nuclei. This marker is called senescence-associated heterochromatin foci (SAHF). Although these are all physiological processes, several factors have been identified that may trigger premature, accelerated senescence. These triggering factors are primarily stress-related, and they are of oxidative, mitochondrial, or endoplasmic origin [26,48]. The response to stressing factors will depend on the level of stress encountered. Low levels of stress induce adaptive responses and upregulate antioxidant capacities and cell turnover. Mild stress levels trigger adaptive responses that enhance cell turnover and increase antioxidant capabilities. However, moderate stress levels can impede stem cell function and reduce proliferation, while high stress levels can severely disrupt cell function. In such cases, pro-inflammatory cytokines and antiangiogenic factors are released, which can lead to abnormal placentation and hastened trophoblast senescence [48].

Oxidative stress damage of the syncytiotrophoblast is associated with mammalian target of rapamycin complex (mTORC1) activation and telomere shortening [49,50,51]. Both of them have been related to the genesis of obstetric complications. Senescent cells present increased levels of mTORC1, a serine-threonine kinase responsible for the induction of anabolism and the inhibition of catabolism by blocking autophagy [39]. The latter is an essential process in the cell recycling system. The inhibition of mTORC1 by rapamycin delays progression into senescence, prevents permanent loss of proliferative capacity, and allows re-entry into the cell cycle of arrest cells [1,39,41].

Critically short telomeres have also been described as a factor that initiates aging [34,40]. They form protective caps at DNA ends, protecting them from breaks and degradation [2,28]. With every consecutive cell division, telomeres progressively shorten and lose their protective ability [2]. Once they reach a critical minimum length, they expose the DNA ends and initiate a DNA response, leading to the activation of the cellular senescence pathway. Telomere shortening has also been associated with environmental stressors such as hyperglycemia, hypoxia, and the aforementioned oxidative stress [32,34,36,41,42].

During the final stages of pregnancy, the decidual cells and placental membranes of the mother also exhibit signs of aging, which may have a significant impact on the signaling pathways necessary for the onset of labor at term [1,3,35,37,52]. Evidence suggests that the expression of certain markers such as p53, p21, IL-6, IL-8, and SA-ß-gal in these tissues is elevated during term labor [53,54]. Pathological early secretion of senescence inflammatory signals (IL-1beta, IL-6, and IL-8) has been observed in premature rupture of membranes (PROM) and preterm birth (PTB) [52,55,56].

This systematic review provides a comprehensive and rigorous synthesis of evidence. We have included both original research and available narrative reviews and assessed the scientific value of both using adequate tools for analysis. Systematic reviews are dependent on the quality and availability of primary studies, and biases and limitations within those studies can affect the reliability of the review findings. Future research will need to include further analysis of the triggering factors, the time of their incidence, and the resulting impact of the degree of placental aging on pregnancy outcomes. To achieve this, a more systematic approach to defining both study and control groups needs to be incorporated into the research.

Our systematic literature review shows that the most significant limitation of the analyzed studies is the definition of fetal growth restriction and differentiating this from SGA. Late FGR and SGA are by definition growth abnormalities that develop after 32 weeks of gestation [5].

The only study that specifically made this differentiation is Paules et al. In their research, both SGA and FGR pregnancies presented signs of accelerated placental senescence, including lower telomerase activity, shorter telomeres, and reduced *SIRT1* RNA expression together with increased *P53* RNA expression [3,26,33]. The FGR cases showed signs of apoptosis, with increased levels of *CASP3* RNA, expression of *SIRT1* RNA, telomerase activity, and telomere length [3,26,33,37]. The caspase-3 activity showed a significant linear increase as the condition’s severity worsened [3]. In this study, both the control and study groups had an ultrasound assessment at 32 weeks, and Doppler parameters were included in the feto-maternal characteristics section of the study. The control group was well-defined but lacked information on the number of spontaneous vaginal deliveries, inductions of labor, or planned cesarean section. Since placental aging has a role in labor initiation, this is critical information for interpreting the results [2,41,42,43].

The largest studied cohort of unexplained stillbirths was published by Ferrari et al. The study group was well-defined, and an initial workup of stillbirth etiologies was performed. Stillbirths were stratified according to birth weight, but only AGA and SGA were defined [28]. Previous studies suggest that large for gestational age is also a risk factor for stillbirth. This would have been an interesting addition to the study [57]. The stillbirths were also stratified as having occurred before and after 34 weeks [28]. It seems a logical strategy to differentiate between unexplained stillbirths related to early vs. late placentation or placental development abnormalities, as this has been done in the study of another placentation-related pathology, preeclampsia [58]. Unfortunately, there is no consensus regarding research on unexplained stillbirths [11,42,46,59]. By definition, stillbirth is death after 22 weeks of gestation or a neonatal weight of 500 g, but WHO recommends reporting only cases after 28 weeks and weight above 1000 g [42]. ACOG recommends stratification of the risk of recurrence and management depending on whether the previous stillbirth occurred before or after 32 weeks of gestation [59]. Many risk factors may contribute to stillbirth [46,59]. The biological pathways are unclear, making it difficult to identify pregnancies with a potential high-risk status or plan an appropriate intervention to reduce the risk [60]. Ferrari et al. tested placenta telomere length reduction as a surrogate for placental premature senescence [28]. They reported that unexplained stillbirth is associated with placental telomere attrition. Interestingly they had two control groups of premature and term deliveries. Premature deliveries included deliveries related to premature rupture of membranes. Telomere lengths in preterm PROM (pPROM) were shorter than in preterm birth with intact membranes and mimicked those present in stillbirth cases, which suggests a possible common pathophysiologic pathway of stillbirth pPROM [1,28].

Indeed, the studies described above specifically show the importance of differentiating between pathological and physiological senescence [40,41,42,43]. They also show that there is a spectrum of abnormal placental aging, which progresses with the severity of the disease, as we hypothesized at the beginning of this review. The delineation between the seriousness of the discussed pathologies could depend on the type and dose of risk exposure and the extent of oxidative stress [28].

## 5. Conclusions

The review revealed an inconsistency in definitions related to abnormal fetal growth, as well as in research methodologies. However, several original studies have identified mechanisms that contribute to abnormal growth. These mechanisms include decreased placental AURK expression with gestational age, increased mtDNA copy number in FGR and SGA placentas, and decreased activity of mTORC1 and mTORC2 in FGR cases. Additionally, alterations in placental gene expression, mitochondrial function, telomere length, and protein expression may contribute to FGR and placental aging. Future research should implement clear inclusion criteria, differentiate between early and late cases, and have a well-described control group to minimize bias and improve our understanding of the role of placental aging in these conditions.

## Figures and Tables

**Figure 1 biomedicines-11-01785-f001:**
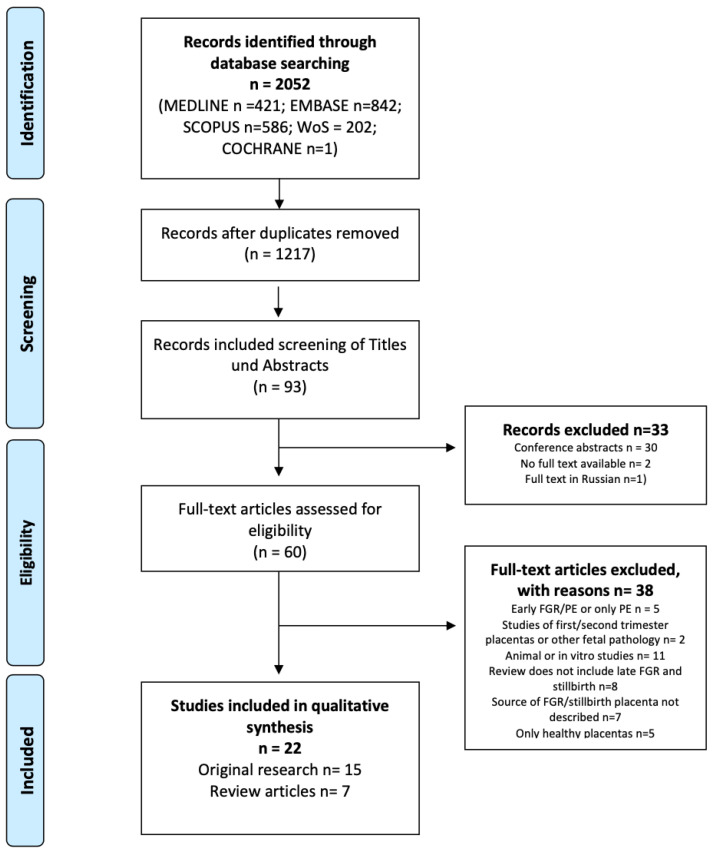
PRISMA flow chart.

**Figure 2 biomedicines-11-01785-f002:**
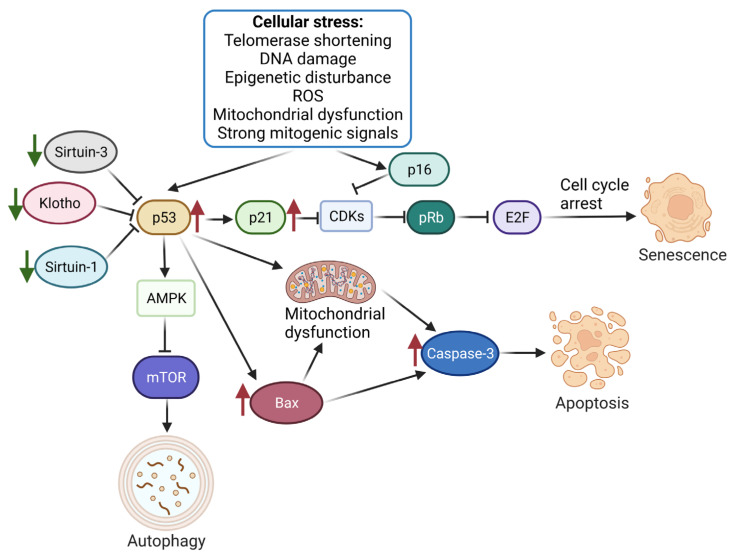
Molecular mechanisms associated with the placental aging process.Cell stress stimulation triggers the activation of the p53/p21 and p16 pathways, leading to CDK inhibition and the subsequent inhibition of Rb phosphorylation. Rb, which binds to E2F to promote DNA replication, is inactivated, resulting in cell replication arrest at the G1/S phase and triggering cellular senescence. Regulatory factors, such as Sirtuin-1/Sirtuin-3 and Klotho, play a role in controlling this process. p53 also stimulates autophagy by regulating AMPK and mTOR. Moreover, the increased expression of p53 during placental aging leads to apoptosis by inducing mitochondrial dysfunction and upregulating the expression of Bax and Caspase-3. Green arrows indicate proteins downregulated during placental aging, while red indicates upregulated proteins. Created with BioRender (24 May 2023). Klotho—Klotho protein involved in aging; Sirtuin-1/Sirtuin-3—proteins responsible for mitochondrial hemostasis; E2F—a group of transcription factors; CDKs—Cyclin-dependent kinases; AMPK—5′ adenosine monophosphate-activated protein kinase.

**Table 1 biomedicines-11-01785-t001:** Search strategy.

Database	Searching Quote
Medline	(“markers of placental aging” OR “cellular senescence” OR “aging” OR “maturation” OR (p53 AND p21) OR (p16 AND pRB) OR “SA-β-gal” OR “mitochondrial dysfunction” OR “mTORC1” OR “telomere shortening” OR “short telomere” OR “telomere attrition”) AND (“placenta” OR “trophoblast” OR “syncytiotrophoblast”) AND (“fetal growth restriction” OR “FGR” OR “intrauterine growth restriction” OR “IUGR” OR “small for gestational age” OR “SGA” OR “stillbirth” OR “Fetal Growth Retardation” [MeSH])
Web of Science	TS = ((“markers of placental aging” OR “cellular senescence” OR “aging” OR “maturation” OR (“p53” AND “p21”) OR (“p16” AND “pRB”) OR “SA-β-gal” OR “mitochondrial dysfunction” OR “mTORC1” OR “telomere shortening” OR “short telomere” OR “telomere attrition”) AND (“placenta” OR “trophoblast” OR “syncytiotrophoblast”) AND (“fetal growth restriction” OR “FGR” OR “intrauterine growth restriction” OR “IUGR” OR “small for gestational age” OR “SGA” OR “stillbirth” OR “Fetal Growth Retardation”))
Scopus and Embase	(TITLE-ABS-KEY(“markers of placental aging”) OR TITLE-ABS-KEY(“cellular senescence”) OR TITLE-ABS-KEY(aging) OR TITLE-ABS-KEY(maturation) OR (TITLE-ABS-KEY(p53) AND TITLE-ABS-KEY(p21)) OR (TITLE-ABS-KEY(p16) AND TITLE-ABS-KEY(pRB)) OR TITLE-ABS-KEY(SA-β-gal) OR TITLE-ABS-KEY(“mitochondrial dysfunction”) OR TITLE-ABS-KEY(mTORC1) OR TITLE-ABS-KEY(“telomere shortening”) OR TITLE-ABS-KEY(“short telomere”) OR TITLE-ABS-KEY(“telomere attrition”)) AND (TITLE-ABS-KEY(placenta) OR TITLE-ABS-KEY(trophoblast) OR TITLE-ABS-KEY(syncytiotrophoblast)) AND (TITLE-ABS-KEY(“fetal growth restriction”) OR TITLE-ABS-KEY(FGR) OR TITLE-ABS-KEY(“intrauterine growth restriction”) OR TITLE-ABS-KEY(IUGR) OR TITLE-ABS-KEY(“small for gestational age”) OR TITLE-ABS-KEY(SGA) OR TITLE-ABS-KEY(stillbirth) OR TITLE-ABS-KEY(“Fetal Growth Retardation”))

**Table 2 biomedicines-11-01785-t002:** Original research studies included in the final synthesis.

Author	Year	Population	Key Points Relevant to Late SGA, FGR, and Stillbirth Research
Beard et al. [25]	2020	13 FGR (8 late FGR) 51 PE 14 PE/FGR 17 term controls	Placental AURK A, B, and C expression decreased with gestational age. Circulating *AURKB* mRNA reduced at term in comparison to pregnancies <34 weeks. AURKC reduced significantly in placentas from pregnancies complicated by severe early-onset FGR. No change in placental *AURKC* expression in FGR at later gestations.
Naha et al. [26]	2020	29 controls 18 FGR 25 SGA	Significant increase in mtDNA copy number in both FGR and SGA compared to control. Sirtuin-3 (Sirt3) protein expression was significantly downregulated in FGR placenta, but there was no significant difference in Nrf1 expression compared to the control group.
Franklin et al. [27]	2019	22 SGA 32 AGA	Alpha klotho levels in cord blood were found to be lower in cases of SGA and maternal vascular malperfusion. These findings suggest a potential link between alpha klotho and the accelerated maturation of placental villi, which is often associated with increased levels of angiopoietin 2. This indicated that alpha klotho may contribute to vascular-mediated placental aging, which can lead to fetal growth restriction.
Maiti et al. [2]	2017	34 GA 37–39 28 late term 4 stillbirth	Aldehyde oxidase 1 and G-protein-coupled estrogen receptor 1 mediate aging of the placenta in stillbirth.
Ferrari et al. [28]	2015	42 stillbirths 43 live term births 15 live preterm births	Telomere length attrition was an essential feature in unexplained SB.
Chen et al. [29]	2015	25 FGR 19 AGA	The activity of mTORC1 and mTORC2 was decreased, and the protein expression of the ubiquitin ligase NEDD4-2 and the ubiquitination of SNAT-2 were increased in FGR placentas. FGR was associated with decreased system A-amino acid transport activity and SNAT-1 and SNAT-2 protein expression in MVM.
Biron-Shental et al. [30]	2014	15 controls 15 FGR	Significant increase in telomere capture in FGR trophoblasts compared to controls. Compensatory response of the FGR placentas to shortened telomeres to maintain homeostasis. Demonstrated the presence of senescence-associated heterochromatin bodies by exposing the involvement of hTERT mRNA and telomere capture mechanisms.
Seidmann et al. [31]	2013	45 term placentas 6 FGR 3 stillbirths	Defective placental maturation was associated with an imbalance of expression of bFGF and PK1.
Paules et al. [3]	2013	21 controls 18 SGA 18 FGR	Both SGA and FGR cases exhibited accelerated placental aging, lower telomerase activity, shorter telomeres, decreased *SIRT1* RNA expression, and increased *P53* RNA expression. In FGR cases, there was also evidence of increased apoptosis, as evidenced by elevated *CASP3* RNA levels. Furthermore, there was a linear trend between the severity of the condition and the levels of *SIRT1* RNA expression, telomerase activity, telomere length, and Caspase-3 activity.
Biron-Shental et al. [32]	2011	5 controls 5 FGR	The *TERC* gene copy number was decreased in FGR trophoblasts.
Heazell et al. [33]	2011	6 normal 6 FGR	In cases of FGR, there was an observed increase in the expression of *P53* mRNA and protein, which were found to be localized in the syncytiotrophoblast. Similar changes were also observed in the expression of *P21* and *BAX*. However, no significant changes were noted in the expression of Mdm2, Bak, and Bcl2. The increased expression of p53 was thought to be associated with altered trophoblast cell turnover, which is similar to the effects of hypoxia exposure.
Biron-Shental et al. [34]	2010	20 controls 14 PE/FGR 14 FGR 9 severe PE/FGR	Telomeres were shorter in PE and FGR placentas. Telomere aggregate formation increased in PE but not FGR. This implied different placental stress-related mechanisms in PE with and without FGR.
Biron-Shental et al. [35]	2010	13 controls 16 FGR	The number and intensity of telomeres staining and telomerase activity were lower in FGR placentas. Telomeres were shorter in trophoblasts of FGR placentas.
Davy et al. [36]	2009	32 FGR 36 controls	Significantly shorter telomeres in FGR placenta samples, but not cord blood samples. There was an association between the suppression of telomerase activity and reduced telomere length in FGR placenta. p21, p16, and EF-1 alpha were significantly elevated in FGR placenta samples.
Kudo et al. [37]	2000	Group A: 31 first trimester chorionic villi specimens of normal pregnancy undergoing abortion Group B: 32 without FGR Group C: 12 with FGR	TA was detected both in CVS and placentas without FGR. Weak TA was observed in FGR. Significantly higher Bcl-2 immunoreactivity was seen in Group A and Group B than in Group C. TUNEL-positive cells were significantly more present in Group C than in Group A or Group B.

AURK—Aurora kinases (A,B,C); mtDNA—mitochondrial DNA; Sirt3—Sirtuin3 protein responsible for mitochondrial hemostasis; Nrf1—transcriptional coactivator responsible for mitochondrial biogenesis; mTOR (C1, C2)—mechanistic target of rapamycin; NEDD4-2—neural precursor cell expressed developmentally down-regulated protein 4-2; MVM—syncytiotrophoblast microvillous plasma membrane; SNAT—sodium-coupled amino acid transporter; hTERT; mRNA—messenger ribonucleic acid; PK1—prokineticin 1; bFGF—basic-fibroblast growth factor; RNA—ribonucleic acid; *TERC*—telomerase RNA component gene; P53/p53—tumor mRNA/protein 52; p21—cyclin-dependent kinase inhibitor; Mdm2—ubiquitin-protein ligase; Bak—mitochondrial single-pass membrane protein; Bcl-2—apoptosis regulator protein; p16—cyclin-dependent kinase inhibitor; EF-1 alpha—elongation factor 1-alpha; TUNEL—terminal d-UTP nick-end labeling; TA—telomerase activity; PE—preeclampsia, FGR—fetal growth restriction, SGA—small for gestational age; CVS—chorionic villi samples.

**Table 3 biomedicines-11-01785-t003:** Review articles included in the final synthesis.

Year	Author	Type of Review	SANRA Quality Assessment	Key Points Relevant to Late SGA, FGR, and Stillbirth Research
2020	Fabiana et al. [38]	Narrative Review	6/12	Telomere erosion in the placenta, with concomitant cellular senescence, may have contributed to SB. Studies that compared telomere length in FGR vs. aged-matched control placentas observed shorter telomeres in FGR placentas. No difference in *TERC* copy numbers between FGR and control, despite reduced telomere length in FGR placentas.
2017	Silva et al. [39]	Narrative Review	8/12	mTOR folate sensing linked folate availability and cell function. Placental mTOR was blocked when there were low maternal folate concentrations and caused a decrease in placental amino acid transport, protein synthesis, and mitochondrial respiration. Fetal growth restriction resulted from the unavailability of nutrients.
2017	Sultana et al. [1]	Expert Review	6/12	FGR coexisted with increased expression of DNA damage biomarkers, reduction of TL and TA, upregulation of p53 and p16, and elevated levels of senescence-associated secretory phenotype and SAHF. Late gestational tissues also showed evidence of placental oxidative DNA damage and premature senescence.
2016	Biron-Shental et al. [40]	Narrative Review	6/12	Telomere homeostasis should be compared in early vs. late-onset FGR. Potential benefits may show oxygen supplementation. Studies of cultured human trophoblasts might help with understanding the mechanisms of placental injury.
2016	Sultana et al. [41]	Narrative Review	6/12	A significantly shorter telomere and/or absent or reduced telomerase activity was observed in the placentas from FGR pregnancies with a reduced expression of *hTERT*. The expression of telomere-induced senescence markers *P21* and *P16* was elevated, and the anti-apoptotic protein Bcl-2 was decreased. OS led to placental insufficiency and an inability to meet the growing fetal demands, leading to death.
2013	Smith et al. [42]	Narrative Review	6/12	Unexplained antepartum stillbirths occurred as a consequence of changes in the late gestation placenta. Small numbers of pregnancies continued beyond 40 weeks; therefore, the adverse effects of genes that produce aging affected fewer pregnancies, and the polymorphisms in genes that produced these effects remained in the population.
2008	Heazell et al. [43]	Narrative review	10/12	Increased apoptosis rate resulting from the activation of the extrinsic or intrinsic apoptotic pathways. Activation of the apoptotic pathway caused an alteration in downstream effector proteins, and specific post-translational modifications/evaluation of oncoprotein expression patterns may have identified candidates that induce apoptosis, such as placental hypoxia, reactive oxygen species, or a reduction in growth factors. Understanding cell turnover pathways may have shown molecular targets capable of inhibiting apoptosis and providing a novel therapeutic approach.

hTERT—human telomerase reverse transcriptase; OS—oxidative stress; Bcl-2—apoptosis regulator protein; p16—cyclin-dependent kinase inhibitor; p21—cyclin-dependent kinase inhibitor; TA—telomere activity, TL—telomere length.

**Table 4 biomedicines-11-01785-t004:** Newcastle—Ottawa quality assessment scale for the observational studies.

	Selection			Comparability (* or **) ^d^	Assessment of Outcome (⋆) ^e^	Total (6*)
	Representativeness of Exposed Cohort (⋆) ^a^	Selection of Non-Exposed Cohort (⋆) ^b^	Ascertainment of Exposure (⋆) ^c^			
Beard et al. [25]	*	*	*	**	*	****** (6)
Naha et al. [26]	-	*	*	*	*	***** (5)
Franklin et al. [27]	*	*	-	**	*	***** (5)
Maiti et al. [2]	*	*	*	**	*	****** (6)
Ferrari et al. [28]	*	*	*	*-	*	***** (5)
Chen et al. [29]	*	*	*	**	*	****** (6)
Biron-Shental et al. [30]	*	*	*	*-	*	***** (5)
Seidmann et al. [31]	*	*	*	**	*	****** (6)
Paules et al. [3]	*	*	*	**	*	****** (6)
Biron-Shental et al. [32]		*	*	*	*	**** (4)
Heazell et al. [33]	*	*	*	*-	*	***** (5)
Biron-Shental et al. [34]	*	*	*	**	*	****** (6)
Biron-Shental et al. [35]		*	*	**	*	***** (5)
Davy et al. [36]		*	*	*	*	**** (4)
Kudo et al. [37]			*	*	*?	*** (3)

Star awarded when: (⋆) ^a^ exposed cohort was chosen from full population or study using nested case-control, (⋆) ^b^ same setting as the exposed cohort, (⋆) ^c^ secure records or directly measured; (⋆) ^d^ one star: excluded or adjusted for prior outcome in analysis, two stars: adjusted for other important variables; (⋆) ^e^ secure records or directly measured. ?—lack of detailed information

## Data Availability

Not applicable.

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
