# Peer review of "Evidence of Placental Aging in Late SGA, Fetal Growth Restriction and Stillbirth—A Systematic Review"

_biomedicines, 2023, doi:10.3390/biomedicines11071785_

Round 1

Reviewer 1 Report

The authors present a systematic analysis of evidence gathered about SGA, fetal growth restriction and stillbirth in literature. This research invetigates the factors related with SGA, fetal growth restriction and stillbirth in late pregnancy.  The topic of research is an original and popular one which is relevant with the field of perinatology as it addresses a very frequently encountered yet desperately and empirically managed issue. The similar research in literature mainly focuses on biochemical and genetic markers for predicting late SGA, fetal growth restriction and stillbirth. However, the this manuscript aims to specify the mechanisms that underlie and connect these markers. The methodology of the study is restricted by the lack of standardization in the conception and execution of the included studies and, thus, related heterogeneity and possible bias. Therefore, the methodology  has been set up as efficiently as possible. The conclusions comply with the evidence presented by the statistical findings and both the conclusions and evidence are all related with the main proposal and conclusion of the manuscript. All of the references are relevant. Although some references date back before 2010, these references are needed as they include theoretical data about the hypothesis of the study. Both the tables and figures are well drawn and demonstrate the required data. In conclusion, I would recommend that the manuscript in its current version can be accepted for publication in Biomedicines.  

The authors present a systematic analysis of evidence gathered about SGA, fetal growth restriction and stillbirth in literature. This research invetigates the factors related with SGA, fetal growth restriction and stillbirth in late pregnancy.  The topic of research is an original and popular one which is relevant with the field of perinatology as it addresses a very frequently encountered yet desperately and empirically managed issue. The similar research in literature mainly focuses on biochemical and genetic markers for predicting late SGA, fetal growth restriction and stillbirth. However, the this manuscript aims to specify the mechanisms that underlie and connect these markers. The methodology of the study is restricted by the lack of standardization in the conception and execution of the included studies and, thus, related heterogeneity and possible bias. Therefore, the methodology  has been set up as efficiently as possible. The conclusions comply with the evidence presented by the statistical findings and both the conclusions and evidence are all related with the main proposal and conclusion of the manuscript. All of the references are relevant. Although some references date back before 2010, these references are needed as they include theoretical data about the hypothesis of the study. Both the tables and figures are well drawn and demonstrate the required data. In conclusion, I would recommend that the manuscript in its current version can be accepted for publication in Biomedicines.  

Author Response

Thank you very much for reviewing our manuscript and the comment you provided. We agree that papers prior to 2010 may be outdated, but (as noted) we wanted to include in our review all papers published to date on the given topic. Hence, we also included papers from before 2010 in our review.

Reviewer 2 Report

The review is important for prraciniants to understand what effect on placenta aging

The senescence process (at morphological and molecular levels) should be described including its relation to hormone action (also in other endocrine organs if possible)

Provide information on the time period the study was performed

If it is possible please add a schematic drawing summarizing what is currently known about placenta aging including senescence

Author Response

Thank you very much for reviewing our manuscript and the comment you provided. We improved our manuscript according to your suggestion. Please see our point-by point answer below as well as new version of manuscript

The review is important for prraciniants to understand what effect on placenta aging

The senescence process (at morphological and molecular levels) should be described including its relation to hormone action (also in other endocrine organs if possible)

Provide information on the time period the study was performed

We have added this information in the material and method section:The retrieved records were downloaded and imported into Zotero's virtual library, which was accessed by the evaluation researchers. After the selection of articles for full-text analysis, they were also deposited in Zotero. The process of synthesizing the results was done using text documents stored in protected files in the virtual workspace. The study was completed on 31.03.2023.”

If it is possible please add a schematic drawing summarizing what is currently known about placenta aging including senescence

We appreciate this suggestion and have included figure 1 and its legend accordingly.

Figure 2. Molecular mechanisms associated with the placental aging process. Cell stress stimulation triggers the activation of the p53/p21 and p16 pathways, leading to CDKs inhibition and subsequent inhibition of Rb phosphorylation. Rb, which binds to E2F to promote DNA replication, is inactivated, resulting in cell replication arrest at the G1/S phase and triggering cellular senescence. Regulatory factors such as SIRTs and Klotho play roles in controlling this process. p53 also stimulates autophagy by regulating AMPK and mTOR. Moreover, the increased expression of p53 during placental aging leads to apoptosis by inducing mitochondrial dysfunction and upregulating the expression of BAX and CASP3. Green arrows indicated proteins downregulated during placental aging while red upregulated. Created with BioRender.

Reviewer 3 Report

The systematic review manuscript is well-written and has presented a critical view of the topic. However, here are the comments needed to be considered for further processing: 

1. Introduction: Please elaborate on the existing reviews on the topic and explain how this work will add something new. The aim of the current study should follow that section.

2. Please indicate the strengths and limitations of the study and provide further directions where more studies are required to do in the future. 

Author Response

Thank you very much for reviewing our manuscript and the comment you provided. We improved our manuscript according to your suggestion. Please see our point-by point answer below as well as new version of manuscript

The systematic review manuscript is well-written and has presented a critical view of the topic. However, here are the comments needed to be considered for further processing: 

  1. Introduction: Please elaborate on the existing reviews on the topic and explain how this work will add something new. The aim of the current study should follow that section.

This has been included in the results section: “The reviews highlight various mechanisms that may contribute to abnormal growth, Telomere erosion and cellular senescence in the placenta were found to contribute to FGR, with shorter telomeres observed in FGR placentas compared to controls. Placental mTOR is blocked when maternal folate concentrations are low, leading to decreased placental amino acid transport and fetal nutrient unavailability. FGR coexists with increased expression of DNA damage biomarkers, reduction of telomere length and telomerase activity, upregulation of senescence-associated markers, and oxidative DNA damage. Placental apoptosis may result from placental hypoxia, reactive oxygen species, or a reduction of growth factors, and understanding cell turnover pathways may provide a novel therapeutic approach. Finally, changes in the late gestation placenta contribute to unexplained antepartum stillbirths, and genes that produce aging affect fewer pregnancies, with polymorphisms in genes that produce these effects remaining in the population.”

The aim of this study is stated in the introduction: “This review aims to present the current understanding of the role of placental aging in late SGA, fetal growth restriction, and term stillbirth and identify future research directions in this area.”

This is the first systematic review that includes solely this scientific question. In that aspect it provides a compendium of knowledge on the topic of the role of placental ageing in abnormal growth leading to stillbirh.

This is adressed in the beginning of the discussion: „The presented research is the first that addresses the problem of late SGA, FGR, and stillbirth in a systematic review. SGA and FGR are two primary forms of prenatally diagnosed abnormal growth. [3] FGR is characterized by alterations in fetoplacental Doppler and poses a greater risk of intrauterine deterioration and mortality compared to SGA. SGA fetuses are often referred to as constitutionally small and have a near-normal perinatal outcome. [3,4] Although both SGA and FGR are associated with suboptimal neurodevelopmental outcomes and intrauterine cardiovascular programming. [41] Placental dysfunction has been most commonly associated with abnormal fetal growth, but only 25% of pregnancies complicated by FGR show abnormalities on histopathological examinations in late forms. [3] For this reason, new approaches are researched to detect subtle changes in placental development.”

  1. Please indicate the strengths and limitations of the study and provide further directions where more studies are required to do in the future.This is adressed in the discussion – edited in line 154

This systematic reviews provides a comprehensive and rigorous synthesis of evidence. We have included both original research and available narrative reviews and assessed the scientific value of both using adequate tools for analysis.  Systematic reviews are dependent on the quality and availability of primary studies, and biases and limitations within those studies can affect the reliability of the review findings. Our systematic literature review has shown that the most significant limitation of the analyzed studies was the definition of fetal growth restriction and differentiating it from SGA. Late FGR and SGA is by definition a growth abnormality that develops after 32 weeks of gestation. [5]

[…]

Indeed, the studies described above specifically have shown the importance of differentiating between pathological and physiological senescence. [37,38,38,40] They also have shown that there is a spectrum of abnormal placental aging, and it progresses with the severity of the disease, as we hypothesized at the beginning of this review. The delineation between the seriousness of the discussed pathologies could depend on the type and dose of risk exposure and the extent of oxidative stress. [25]

Round 2

Reviewer 2 Report

I am partially happy with the corrections

No answer to the question below was provided. Short paragraph on this issue for readers who are not familiar with this process need to be added

"The senescence process (at morphological and molecular levels) should be described including its relation to hormone action (also in other endocrine organs if possible)"

Author Response

Dear Reviewer,

We have added more information to the manuscript about the mechanism as suggested: 

At the morphological level, senescence is associated with several characteristic features. These include cellular and tissue atrophy, reduced cell proliferation, altered cell morphology, and the accumulation of cellular debris. In many tissues, such as the skin, there is a decline in the number and activity of specialized cells like fibroblasts and melanocytes, leading to the appearance of wrinkles, thinning, and loss of elasticity. [17,18]

At the molecular level, senescence involves various mechanisms, including alterations in gene expression, telomere shortening, genomic instability, epigenetic modifications, mitochondrial dysfunction, and cellular senescence-associated secretory phenotype (SASP). Hormones play a crucial role in modulating these processes and influencing the pace of senescence. [19]

One of the well-studied hormonal systems related to senescence is the hypothalamic-pituitary-adrenal (HPA) axis. The HPA axis regulates the production and release of cortisol, a hormone involved in stress response. With age, the HPA axis becomes dysregulated, leading to altered cortisol levels. Elevated cortisol levels can accelerate the aging process, affecting multiple organ systems and contributing to the development of age-related diseases. [20,21]

We hope, our manuscript is complete now and can be published.

Best regards

Anna Kajdy MD, PhD